# Effect of neostigmine/glycopyrrolate versus sugammadex on postoperative delirium in older adults: A triple-masked, randomized, controlled trial protocol

Wei Dou[1,2☯], Kai Jiang[1,2☯], Jing-hui Hu[1,2☯], Min-yuan Zhuang[1,2], Hong Liu[3], Fu-hai Ji[1,2], Ke Peng[1,2]*

1 Department of Anesthesiology, The First Affiliated Hospital of Soochow University, Suzhou, Jiangsu, China, 2 Institute of Anesthesiology, Soochow University, Suzhou, Jiangsu, China, 3 Department of Anesthesiology and Pain Medicine, University of California Davis Health, Sacramento, California, United States of America

☯ These authors contributed equally to this work.
* pengke0422@163.com

## Abstract

### Background

Postoperative delirium (POD) is an acute disturbance of attention and awareness in older adults undergoing surgery. It is associated with prolonged hospital stay and increased morbidity and mortality. Recent studies suggested that neostigmine, an acetylcholinesterase inhibitor used to reverse neuromuscular blockade, may reduce POD risk. We aim to evaluate whether neuromuscular blockade reversal with neostigmine/glycopyrrolate reduces POD compared with sugammadex.

### Methods

This single-center, triple-masked, randomized, controlled superiority trial will enroll 320 older adults scheduled for major non-cardiac and non-neurosurgical surgery. Patients will be randomized (1:1) to receive either neostigmine 40 µg/kg plus glycopyrrolate 8 µg/kg or sugammadex 2 mg/kg for neuromuscular blockade reversal at the end of surgery. The primary outcome is the incidence of POD within postoperative 7 days or until discharge, assessed twice daily with the validated Chinese version of 3-min Diagnostic Interview for Confusion Assessment Method. Secondary outcomes include days with POD and proportion of hospital days affected; POD severity assessed using the highest score and the sum scores of Confusion Assessment Method Severity; and 30-day cognitive function assessed using the 10-item Telephone Interview of Cognitive Status.

**Data availability statement:** No datasets were generated or analysed during the current study. All relevant data from this study will be made available upon study completion.

**Funding:** This study will be funded by the National Natural Science Foundation of China (82471290) and Prof. Changgeng Ruan's Research and Innovation Fund for Graduate Students, the First Affiliated Hospital of Soochow University (RSJCX202408). The funders had no role in study design, data collection and analysis, decision to publish, or preparation of the manuscript.

**Competing interests:** The authors have declared that no competing interests exist.

## Discussion

Results of this study will determine whether neostigmine/glycopyrrolate offers a simple, low-cost strategy to prevent POD and will inform evidence-based selection of neuromuscular reversal agents in older surgical patients.

## Trial registration

Chinese Clinical Trial Registry (ChiCTR2400093158).

---

## Introduction

Postoperative delirium (POD) is an acute, fluctuating disturbance of attention and awareness that typically emerges within seven days after surgery, affecting 20–30% of older adults [1–3]. It prolongs hospital stay, increases falls, and is associated with increased morbidity and mortality. POD arises from a complex interplay of predisposing factors (advanced age, pre-existing cognitive impairment) and perioperative insults (systemic inflammation, oxidative stress, and impaired cholinergic neurotransmission) [4,5].

Neuromuscular blockade is routinely reversed at the end of general anesthesia. Neostigmine, an acetylcholinesterase inhibitor, augments synaptic acetylcholine and is usually combined with glycopyrrolate to limit muscarinic side-effects. Although neostigmine does not readily cross the blood-brain barrier, preliminary data suggest it may reduce cognitive dysfunction via peripheral anti-inflammatory or anti-oxidative pathways linked to enhanced cholinergic signaling [6,7], or through a cerebral vascular mechanism [8]. Sugammadex, a selective relaxant-binding agent, provides faster reversal with fewer autonomic effects and no direct cholinergic activity [9].

Available evidence regarding neostigmine and postoperative cognition remains inconsistent: small studies have indicated that neostigmine may attenuate early postoperative cognitive dysfunction [6,10], whereas others found no benefit over sugammadex [11,12]. We therefore designed this randomized clinical trial to test whether neostigmine/glycopyrrolate reversal decreases POD incidence compared with sugammadex in older patients undergoing major non-cardiac surgery.

## Methods

### Trial design

This is a single-center, randomized, triple-masked, controlled superiority trial. The protocol was approved by the Ethics Committee of the First Affiliated Hospital of Soochow University (2024−477) on November 13, 2024 and registered at the Chinese Clinical Trial Registry (ChiCTR2400093158) on November 29, 2024. The study will adhere to the Declaration of Helsinki. Written informed consent will be obtained from all patients. The protocol follows the Standard Protocol Items: Recommendations for Interventional Trials (SPIRIT) 2025 statement [13].

Currently, this trial is recruiting patients; the recruitment started on December 1, 2024 and is anticipated to end in May 2026. Data collection will be completed 30

days after the surgery for the last recruited patient. Study results are expected to be published in a peer-reviewed journal in 2027. **Fig 1** is the planned patient enrolment, study interventions, and assessment timeline, structured in line with the SPIRIT guidelines. The trial flow diagram is shown in **Fig 2**.

### Eligibility criteria

To be included in this study, patients should meet the following criteria: (1) age ≥ 65 years; (2) ASA classifications I–III; (3) scheduled for major non-cardiac and non-neurosurgical surgery (thoracic, abdominal, urologic, orthopedic, and spinal procedures) expected to last ≥ 1.5 hours); and (4) planned postoperative ward stay ≥ 2 days.

| | STUDY PERIOD | | | | | | | |
|---|---|---|---|---|---|---|---|---|
| | Enrollment | Allocation | Post-allocation | | | | | Close-out |
| **TIMEPOINT** | Preanesthetic visit | After induction | End of surgery | PACU | 24 h | 48 h | 3–7 days | 30 days |
| **ENROLMENT** | | | | | | | | |
| Eligibility screening | × | | | | | | | |
| Written informed consent | × | | | | | | | |
| Baseline data | × | | | | | | | |
| MMSE score | × | | | | | | | |
| Randomization | | × | | | | | | |
| Allocation | | × | | | | | | |
| **INTERVENTIONS** | | | | | | | | |
| Neostigmine/glycopyrrolate | | | × | | | | | |
| Sugammadex | | | × | | | | | |
| **ASSESSMENTS** | | | | | | | | |
| Incidence of POD | | | | | × | × | × | |
| Days with POD | | | | | × | × | × | |
| Percentage of POD | | | | | × | × | × | |
| Peak CAM-S score | | | | | × | × | × | |
| Sum of CAM-S scores | | | | | × | × | × | |
| 30-d cognitive function | | | | | | | | × |
| Pain at rest | | | | | × | × | | |
| Pain on movement | | | | | × | × | | |
| Use of PCIA | | | | | × | × | | |
| Rescue analgesic use | | | | × | × | × | × | |
| Quality of recovery | | | | | × | × | | |
| Incidence of PONV | | | | | × | × | | |
| Non-delirium complications [a] | | | | × | × | × | × | × |
| 30-d mortality | | | | | | | | × |

**Fig 1. Schedule of enrollment, interventions, and assessments.** Notes: POD, postoperative delirium; PCIA, patient-controlled intravenous analgesia; PACU, post-anesthesia care unit; PONV, postoperative nausea and vomiting. [a] Including hypoxemia, pulmonary edema, pulmonary infection, respiratory failure, myocardial infarction, new-onset atrial fibrillation, heart failure, gastrointestinal bleeding, stroke, renal failure, hemorrhagic shock, sepsis, septic shock, anastomotic leak, and reoperation.

 

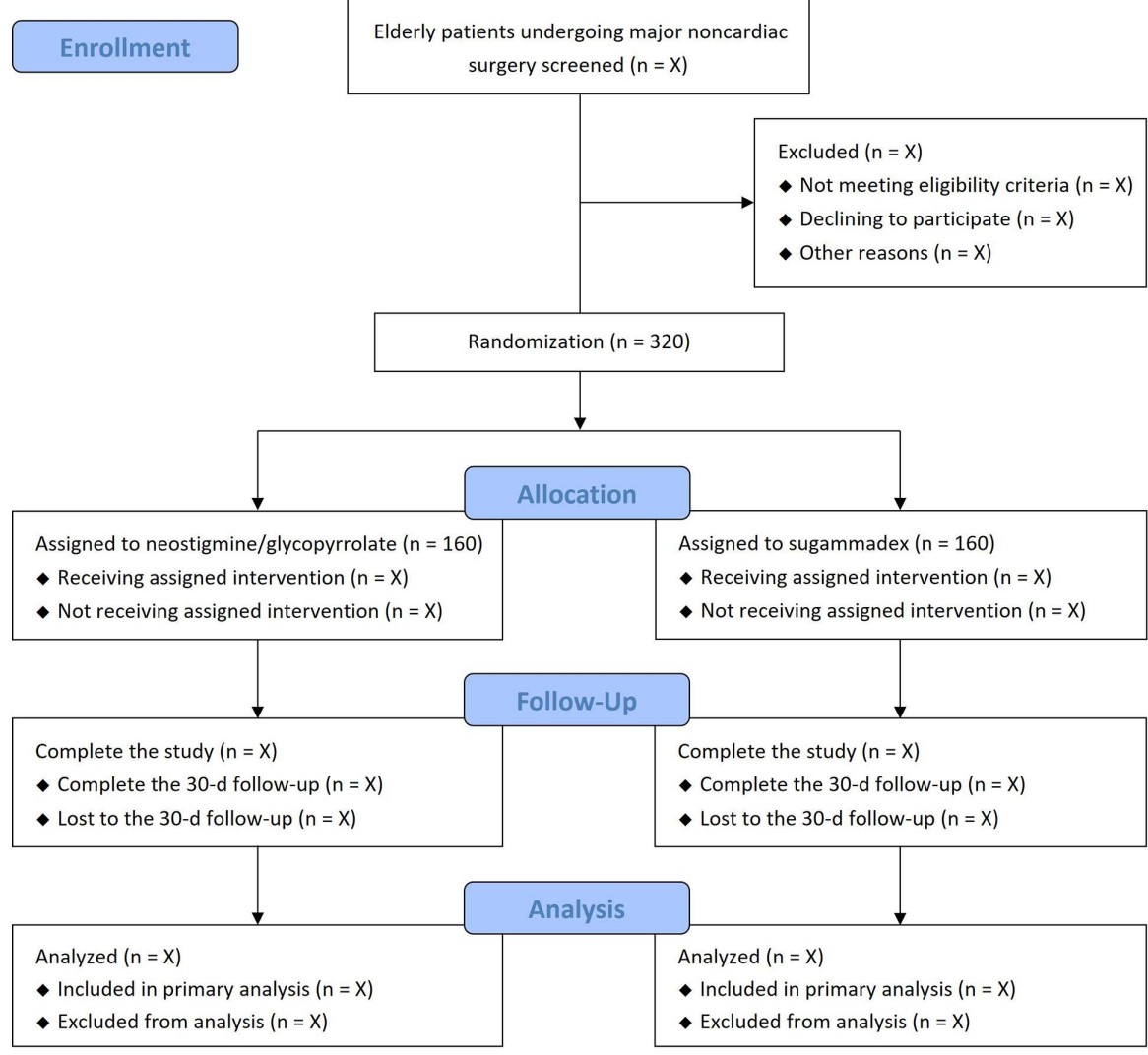

**Fig 2. Trial flow diagram.**

Exclusion criteria are: (1) known allergy to study drugs; (2) contraindications to neostigmine (epilepsy, intestinal obstruction, urinary tract obstruction, asthma, glaucoma), paralysis, or neuromuscular diseases; (3) Chil-Pugh C liver impairment or renal failure; (4) chronic use of cholinesterase inhibitors, anticholinergics, or psychotropics; (5) inability to communicate or refusal to consent.

## Randomization and masking

An independent researcher generated a computerized block randomization (1:1; blocks of 2 and 4) using sealedenvelope. com. Allocation codes are sealed in sequentially numbered, opaque envelopes that will be opened by a research nurse after induction of anesthesia. Patients will be randomized to one of the two study groups: the neostigmine/glycopyrrolate group or the sugammadex group. Investigational drugs are reconstituted in identical 10 mL syringes with 0.9% saline. All patients, anesthesia staff, outcome assessors, and a statistician will be masked to allocation.

## Anesthetic care

Standard monitoring includes electrocardiography, pulse oximetry, non-invasive blood pressure, train of four (TOF), and bispectral index (BIS). TOF monitoring will be performed using TOF-Watch SX at the adductor pollicis muscle via ulnar nerve stimulation at the wrist. We will follow a prespecified calibration protocol: after induction and before rocuronium administration, the device is calibrated to achieve supramaximal stimulation and a stable baseline TOF ratio > 0.9; calibration will be repeated until three consecutive readings vary by < 5%. Our quantitative neuromuscular monitoring protocol adheres to current international recommendations for quantitative monitoring (ESAIC 2023 and ASA 2023 guidelines) [14,15], ensuring standardized calibration and assessment of neuromuscular block and recovery.

Anesthesia will be induced with intravenous sufentanil 0.3 µg/kg, propofol 1.5–2 mg/kg, and rocuronium 0.6 mg/kg. Following tracheal intubation, patients will receive sevoflurane 1–3% inhalation to achieve an appropriate anesthesia depth (BIS 40–60). Remifentanil will be intravenously infused for intraoperative anti-nociception. Additional rocuronium 0.15 mg/kg will be given to maintain TOF counts 1–2. Patients will receive dexamethasone 5 mg during induction and flurbiprofen axetil 50 mg and palonosetron 0.075 mg during surgery. Postoperative pain will be treated with patient-controlled intravenous analgesia (PCIA), containing sufentanil 100 µg diluted with 0.9% saline to 100 mL. Pain intensity will be assessed using an 11-point numeric rating scale (NRS), with 0 denoting no pain and 10 denoting the most severe pain. If the NRS pain score is ≥ 4, rescue analgesia with flurbiprofen axetil or opioids will be given.

## Study interventions

When surgery is completed and TOF count ≥ 3 is reached, patients in the neostigmine/glycopyrrolate group will receive neostigmine 40 µg/kg plus glycopyrrolate 8 µg/kg intravenously, whereas those in the sugammadex group will receive intravenous sugammadex 2 mg/kg intravenously [6,16]. Patients will be sedated with propofol infusion of 1–3 mg/kg/h to achieve a BIS of 60–80 until TOF ratio > 0.9. Extubation is performed when TOF ratio ≥ 0.9, consciousness is regained, and airway reflexes have returned. Patients will then be transferred to the post-anesthesia care unit (PACU).

## Study outcomes

The primary outcome is the incidence of POD within postoperative 7 days or until discharge, whichever occurs first. Trained researchers who are masked to group assignment will use the validated Chinese version of 3-min Diagnostic Interview for Confusion Assessment Method (3D-CAM) to assess POD twice daily (08:00–10:00 and 19:00–21:00) [17,18]. When characteristics 1 (acute onset) and 2 (inattention) are present simultaneously, along with either characteristic 3 (disorganized thinking) or 4 (altered consciousness), the clinical diagnosis of POD is established [19,20]. In addition to the twice-daily POD assessments, a 12-h look-back data extraction from medical records and nursing notes will be performed to capture any CAM-positive signs that may have occurred within the preceding 12 hours [21]. The day of the first positive 3D-CAM assessment (morning, evening, or 12-h look-back) is taken as the event date.

Secondary outcomes: (1) days with POD and proportion of hospital days affected; (2) POD severity, assessed using the highest score and the sum scores of Confusion Assessment Method Severity (CAM-S) [22]; and (3) 30-day cognitive function, assessed using the 10-item version of the Telephone Interview of Cognitive Status (TICS-10) comprising time orientation (awareness of year, month, day, day of week, and season) and mathematical calculations (serial subtracting 7 from 100, up to 5 times) [23].

Exploratory outcomes: (1) 24-h and 48-h NRS pain scores at rest and on movement; (2) PCIA consumption and rescue analgesia (0–24 h and 24–48 h); (3) quality of recovery at 24 h and 48 h, assessed using the 15-item Quality of Recovery Scale (QoR-15) [24,25]; (4) postoperative nausea and vomiting during 0–48 h postoperatively; (5) non-delirium complications (hypoxemia, pulmonary edema, pulmonary infection, respiratory failure, myocardial infarction, new-onset atrial fibrillation, heart failure, gastrointestinal bleeding, stroke, renal failure, hemorrhagic shock, sepsis, septic shock, anastomotic

leak, and reoperation); (6) length of postoperative hospital stay; and (7) 30-day mortality. In addition, the neuromuscular recovery outcomes include time to TOF ratio > 0.9, incidence of residual neuromuscular blockade at extubation (TOF ratio < 0.9), time to extubation, and length of PACU stay.

## Data collection and management

The day before or on the morning of surgery, investigators blinded to allocation will conduct the preanesthetic visit on the ward to screen for eligibility and record baseline data. Baseline cognitive function will be assessed with the 30-point Chinese version of the Mini-Mental State Examination (MMSE), covering six domains: orientation to time and place, word repetition, attention and calculation, language expression, speech comprehension, and simple motor commands [26,27]. Education-adjusted cutoff values for cognitive impairment will be applied: < 24 for patients with postsecondary education, < 23 for secondary education, < 20 for primary education, and < 18 for illiterate patients.

Intraoperative and surgical data will be retrieved from the electronic medical records and the anesthesia information system. Cumulative propofol dose from study drug administration (reversal) to extubation will be recorded. Postoperative follow-up will be performed by two independent outcome assessors, who will undergo standardized training prior to study initiation (including completion of the online 3D-CAM training and bedside practical training supervised by a board-certified neuropsychologist), as specified in our recent pilot study [20]. Inter-rater reliability will be assessed during the initial phase of the trial (first 20 patients) using Cohen's kappa; an acceptable value of ≥ 0.80 will be required before formal recruitment proceeds.

Data will be entered into designated case report forms and uploaded to a secure electronic database. A statistician blinded to allocation will perform all analyses. The principal investigator (KP) will ensure data integrity. An independent Data Monitoring Committee will oversee trial conduct each year.

## Sample size calculation

The incidence of POD after major non-cardiac surgery in older adults is 20–30% [1,3]. A placebo-controlled trial showed that neostigmine reduced the incidence of early postoperative cognitive dysfunction from 36.2% to 14.3% (absolute difference 22%) [6]. We conservatively assumed a 12.5% absolute risk reduction (25% with sugammadex vs. 12.5% with neostigmine/glycopyrrolate), which is supported by our pilot trial conducted at the same center using identical 3D-CAM assessments [20]. In that study, active interventions showed approximately 11% absolute reductions in POD incidence. This empirical evidence, combined with the anti-inflammatory and anti-oxidative properties of enhanced cholinergic signaling by neostigmine, supports the clinical plausibility of the assumed effect size. With α = 0.05 (two-sided) and 80% power, 152 patients per group are required (PASS 15; NCSS, LLC, Kaysville, UT, USA). Allowing for 5% attrition, we will enroll 320 patients (n = 160 per group). We acknowledge that the assumed 12.5% absolute risk reduction, while supported by our pilot data and mechanistic rationale, may prove optimistic; a smaller true effect would result in reduced statistical power.

## Statistical analysis

Normally distributed continuous variables will be expressed as mean ± standard deviation and compared using unpaired t test; non-normally distributed variables as median (interquartile range) and compared using Mann-Whitney U test. Categorical variables will be summarized as number (percentage) and compared using $\chi^2$ test or Fisher's exact test, as appropriate. Treatment effects will be reported as mean/median differences or risk differences with the 95% confidence intervals (CIs).

The primary analysis will be conducted in a modified intention-to-treat population, defined as all randomized patients who undergo surgery and have at least one postoperative POD assessment. In our recently completed pilot trial using

identical assessment protocols [20], twice-daily 3D-CAM evaluations were completed without exception in all 78 older surgical patients (100% ascertainment). Based on this empirical evidence, we expect minimal missing data (< 5%) and will perform the primary analysis on a complete case basis, assuming that data are missing completely at random. To assess robustness to any unforeseen missingness, we will conduct: (1) worst-case scenario analysis (all missing data assumed to be POD in the neostigmine/glycopyrrolate group and no POD in the sugammadex group); (2) best-case scenario analysis (reverse of worst-case); and (3) per-protocol analysis (excluding patients with major protocol deviations). If the observed missing data rate exceeds 10%, an additional sensitivity analysis using multiple imputation will be conducted.

Kaplan–Meier curves will be used to describe time to first POD, with the day of surgery set as day 0. Time-to-event analysis will be conducted on a discrete-time scale (calendar days), which constitutes an approximation given that the twice-daily POD assessments result in interval-censored event timing within a 12-hour window. The treatment effect will be estimated with a Cox proportional-hazards model. As a sensitivity analysis for competing risks (death and early discharge), we will estimate cumulative incidence functions using the Fine-Gray model. If the proportional hazards assumption is violated (Schoenfeld residuals test $P < 0.05$), we will use a time-dependent covariate model or stratified Cox model. Cumulative propofol exposure will be incorporated as a continuous covariate in the logistic regression model for the primary binary outcome and in the Cox proportional-hazards model for time-to-event analysis, with effect estimates reported as adjusted odds ratio and hazard ratio, respectively.

Subgroup analyses of the POD incidence will be conducted according to age (65–80 years or > 80 years), sex (male or female), ASA classification (I–II or III), preoperative MMSE (< 23 or ≥ 23), and type of surgery (thoracic, abdominal, orthopedic/spinal, urologic). We will perform interaction tests (treatment-by-surgical-type) to assess whether the treatment effect on POD differs significantly across surgical subspecialties. All subgroup and interaction analyses are considered exploratory, and findings will be interpreted cautiously as hypothesis-generating rather than confirmatory. If significant heterogeneity is detected ($P$ for interaction < 0.10), we will report stratum-specific estimates; otherwise, the pooled estimate will be presented as the primary result.

The primary outcome window (within 7 postoperative days or until discharge) may lead to differential outcome ascertainment if discharge timing differs between groups. We will conduct a sensitivity landmark analysis restricted to patients with a minimum observation time of 72 hours (i.e., those remaining hospitalized for ≥ 3 days), ensuring comparable ascertainment windows between groups. In our pilot study, the median postoperative hospital stay was 7–10 days with only 2 patients (< 3%) discharged before postoperative day 3 [20], indicating that this sensitivity analysis will retain > 97% of the cohort while effectively addressing potential ascertainment bias.

To control for multiplicity, we will apply the Bonferroni correction for three secondary outcomes (days with POD, highest CAM-S score, and 30-day cognitive function), using a significance threshold of α = 0.017 (0.05/3). Secondary analyses will serve a supportive role for the primary outcome and help generate hypotheses for future research, with findings interpreted accordingly. All exploratory outcomes (including pain scores, QoR-15 scores, non-delirium complications, 30-day mortality, and neuromuscular recovery metrics) will be analyzed using descriptive statistics only and interpreted as hypothesis-generating without adjustment for multiple comparisons. Statistical analysis will be performed using the GraphPad Prism 9 (Boston, MA, USA) and R software version 4.3. A two-sided $P$ value < indicates statistical significance.

## Discussion

This single-center, triple-masked, clinical trial will randomize 320 older adults undergoing major non-cardiac surgery to receive either neostigmine/glycopyrrolate or sugammadex for neuromuscular blockade reversal. Our primary hypothesis is that the neostigmine regimen would reduce the incidence of POD within seven days compared with sugammadex. The implementation and reporting of this trial will adhere strictly to the CONSORT 2025 statement [28].

The potential mechanisms underlying the cognitive protection of neostigmine are as follows. First, by stimulating the peripheral cholinergic anti-inflammatory pathway, it suppressed systemic and hippocampal IL-1β release and neuronal

loss, thereby mitigating surgery-induced cognitive decline in rats [7]. Second, in older adults undergoing non-cardiac surgery, neostigmine reduced early postoperative cognitive dysfunction, attenuated oxidative stress, and elevated serum brain-derived neurotrophic factor [6]. Third, independent of blood-brain-barrier penetration, it increased cerebral perfusion via peripheral cholinergic activation, a vascular mechanism that alone is sufficient to confer cognitive protection in rodent models [8].

Sugammadex, in contrast, offers no such cholinergic or anti-inflammatory effects; its benefit is limited to rapid and complete neuromuscular reversal, which has not been shown to translate into cognitive advantage. Clinical findings regarding neostigmine and postoperative cognition remain inconsistent, possibly because of heterogeneity in anticholinergic co-medications (atropine or glycopyrrolate) and the absence of quantitative neuromuscular monitoring [6,10–12,29]. Notably, atropine impairs cognition whereas glycopyrrolate does not [30,31]. Consequently, firm evidence to support cholinesterase inhibitors for the prevention or treatment of POD is still lacking [32–34].

Our protocol standardizes glycopyrrolate co-administration and mandates quantitative TOF monitoring, thereby highlighting the action of neostigmine. Other strengths of our design include random allocation, triple masking, validated delirium assessments (3D-CAM twice daily), and pre-specified subgroup analyses. If our hypothesis is confirmed, neostigmine/glycopyrrolate would represent a simple, low-cost intervention to reduce POD in a high-risk population. However, if the true effect size is smaller than the assumed 12.5% absolute risk reduction, or if biological heterogeneity increases variance, the findings may be null or negative. Such a finding would suggest that the observed cholinergic effects of neostigmine do not translate into clinically meaningful cognitive protection in the broad surgical population. Given the biological complexity of POD, it is possible that the treatment effect may be heterogeneous across subgroups, and the overall trial results may mask beneficial effects in specific phenotypes that would warrant investigation in future studies.

Main limitations are the single-center setting, exclusion of cardiac patients and those admitted to the intensive care unit (ICU), and lack of cholinesterase biomarker assays. The exclusion of cardiac surgery patients and patients requiring postoperative ICU admission limits the external validity of our findings to these high-risk populations, where the pathophysiology of delirium may differ (e.g., involving systemic inflammation induced by cardiopulmonary bypass or encephalopathy in critical illness). It is our design choice to isolate the specific effects of reversal agents from these confounding factors. The inclusion of a heterogeneous surgical population (thoracic, abdominal, orthopedic, spinal, and urologic surgeries) may dilute the effect size and introduce clinical heterogeneity that could affect POD incidence; however, we intend to enhance the generalizability of findings across the broad spectrum of major non-cardiac surgery to reflect real-world clinical practice where reversal agents are used uniformly regardless of surgical type. Pre-specified subgroup analyses can mitigate this concern. Our pilot study was also conducted in the same surgical population [20]. Although we anticipate minimal missing data based on our pilot experience, the primary analysis assumes data are missing completely at random. If missingness were related to unobserved factors (e.g., clinical deterioration or early discharge), effect estimates could be biased; this will be addressed through the pre-specified sensitivity analyses (worst-case and best-case scenarios).

In summary, this trial will determine whether neuromuscular blockade reversal with neostigmine combined with glycopyrrolate, compared to sugammadex, can reduce the incidence of POD in older adults undergoing major non-cardiac surgery. The findings will inform evidence-based selection of reversal agents and may offer an inexpensive, readily implementable strategy to protect postoperative cognitive recovery.

## Supporting information

**S1 File. Protocol in original language.**
(PDF)

**S2 File. Protocol in English.**
(DOCX)

**S3 File. SPIRIT checklist.**
(DOCX)

## Author contributions

**Conceptualization:** Wei Dou, Kai Jiang, Jing-hui Hu, Min-yuan Zhuang, Hong Liu, Fu-hai Ji, Ke Peng.

**Funding acquisition:** Min-yuan Zhuang, Ke Peng.

**Investigation:** Wei Dou.

**Methodology:** Wei Dou, Kai Jiang, Jing-hui Hu, Hong Liu, Ke Peng.

**Resources:** Min-yuan Zhuang, Ke Peng.

**Supervision:** Fu-hai Ji, Ke Peng.

**Visualization:** Wei Dou.

**Writing – original draft:** Wei Dou, Kai Jiang, Jing-hui Hu, Min-yuan Zhuang.

**Writing – review & editing:** Hong Liu, Fu-hai Ji, Ke Peng.

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
