## [Decision Letter · Decision Letter 0]

8 Jan 2026

Dear Dr. Peng,

Thank you for submitting your manuscript to PLOS ONE. After careful consideration, we feel that it has merit but does not fully meet PLOS ONE’s publication criteria as it currently stands. Therefore, we invite you to submit a revised version of the manuscript that addresses the points raised during the review process.

This protocol describes a well-designed, triple-masked, randomized trial with a clearly defined primary endpoint. Key methodological concerns requiring clarification before implementation include: the handling of missing outcome data, which currently aligns with a modified intention-to-treat approach and risks bias if missing non-randomly; the detailed statistical specification for the time-to-event analysis of POD, including the handling of interval-censored data and discharges; the potential for differential post-reversal propofol exposure between groups as a confounder; and the risk of differential outcome ascertainment due to variable discharge timing, necessitating pre-specified sensitivity analyses. The sample size calculation, while explicit, would benefit from a stronger justification for the assumed effect size on delirium. The plan should also address multiplicity given the numerous secondary outcomes, specify handling of clinical heterogeneity from the mixed surgical population, and provide more detail on quantitative neuromuscular monitoring. Strengths include standardized anesthesia, rigorous masking, and proper registration. Minor revisions suggested for improved reproducibility and clarity involve detailing assessor training, specifying data sharing plans, and streamlining terminology..

We look forward to receiving your revised manuscript.

Kind regards,

Lalit Gupta

Academic Editor

PLOS One

Journal Requirements:

This study will be funded by the National Natural Science Foundation of China (82471290) and Prof. Changgeng Ruan’s Research and Innovation Fund for Graduate Students, the First Affiliated Hospital of Soochow University (RSJCX202408). The funders have no role in the study design, data collection, data analysis, interpretation, or writing of this manuscript.

Additional Editor Comments:

This protocol describes a well-designed, triple-masked, randomized trial with a clearly defined primary endpoint. Key methodological concerns requiring clarification before implementation include: the handling of missing outcome data, which currently aligns with a modified intention-to-treat approach and risks bias if missing non-randomly; the detailed statistical specification for the time-to-event analysis of POD, including the handling of interval-censored data and discharges; the potential for differential post-reversal propofol exposure between groups as a confounder; and the risk of differential outcome ascertainment due to variable discharge timing, necessitating pre-specified sensitivity analyses. The sample size calculation, while explicit, would benefit from a stronger justification for the assumed effect size on delirium. The plan should also address multiplicity given the numerous secondary outcomes, specify handling of clinical heterogeneity from the mixed surgical population, and provide more detail on quantitative neuromuscular monitoring. Strengths include standardized anesthesia, rigorous masking, and proper registration. Minor revisions suggested for improved reproducibility and clarity involve detailing assessor training, specifying data sharing plans, and streamlining terminology.

Reviewers' comments:

Reviewer's Responses to Questions

**Comments to the Author**

1. Does the manuscript provide a valid rationale for the proposed study, with clearly identified and justified research questions?

Reviewer #1: Yes

Reviewer #2: Yes

2. Is the protocol technically sound and planned in a manner that will lead to a meaningful outcome and allow testing the stated hypotheses?

Reviewer #1: Partly

Reviewer #2: Yes

3. Is the methodology feasible and described in sufficient detail to allow the work to be replicable?

Reviewer #1: Yes

Reviewer #2: Yes

4. Have the authors described where all data underlying the findings will be made available when the study is complete?

The PLOS Data policy requires authors to make all data underlying the findings described in their manuscript fully available without restriction, with rare exception, at the time of publication. The data should be provided as part of the manuscript or its supporting information, or deposited to a public repository. For example, in addition to summary statistics, the data points behind means, medians and variance measures should be available. If there are restrictions on publicly sharing data—e.g. participant privacy or use of data from a third party—those must be specified.requires authors to make all data underlying the findings described in their manuscript fully available without restriction, with rare exception, at the time of publication. The data should be provided as part of the manuscript or its supporting information, or deposited to a public repository. For example, in addition to summary statistics, the data points behind means, medians and variance measures should be available. If there are restrictions on publicly sharing data—e.g. participant privacy or use of data from a third party—those must be specified.

Reviewer #1: Yes

Reviewer #2: Yes

5. Is the manuscript presented in an intelligible fashion and written in standard English?

Reviewer #1: Yes

Reviewer #2: Yes

You may also provide optional suggestions and comments to authors that they might find helpful in planning their study.

Reviewer #1: This manuscript presents a well-structured study protocol for a single-center, triple-masked randomized controlled trial comparing neostigmine/glycopyrrolate with sugammadex for neuromuscular blockade reversal, with postoperative delirium (POD) as the primary outcome in older adults. The clinical rationale is relevant, the research question is clearly stated, and the protocol includes ethical approval and trial registration. Overall, the study has the potential to address an important perioperative question; however, several methodological and statistical aspects require clarification to strengthen internal validity and replicability.

Major comments:

The analysis population is described as including randomized patients with available outcome data, and no imputation is planned for missing data. This corresponds to a modified intention-to-treat approach and may introduce bias if POD assessments are missing non-randomly (e.g., early discharge or clinical deterioration). The authors should clearly define the primary analysis set (ITT vs modified ITT) and pre-specify how missing POD assessments will be handled, including consideration of sensitivity analyses.

The statistical plan mentions Kaplan–Meier analysis and hazard ratios for time to POD, but does not specify the survival model, time origin, or how event time will be assigned given twice-daily assessments (with potential interval censoring). These elements should be explicitly defined, and assumptions (e.g., proportional hazards) should be addressed.

The post-reversal propofol sedation protocol (“until TOF ratio >0.9”) may result in differential sedative exposure between groups, given the faster reversal profile of sugammadex. As sedative exposure is plausibly related to POD risk, the authors should clarify whether propofol dose and duration will be recorded and whether this variable will be adjusted for or examined in sensitivity analyses.

The primary outcome window (“within 7 postoperative days or until discharge”) may lead to differential outcome ascertainment if discharge timing differs between groups. The authors should clarify how this limitation will be addressed analytically and consider sensitivity analyses based on minimum observation time.

Minor comments:

Additional details on assessor training and any assessment of inter-rater reliability for POD diagnosis would improve reproducibility.

Clarification on whether MMSE scores are adjusted for education or language norms would strengthen baseline cognitive characterization.

The data availability statement could be made more specific by indicating the intended repository, de-identification process, and timing of data release.

Addressing these points would improve transparency and reduce analytical flexibility, thereby strengthening the protocol’s ability to test the stated hypothesis.

Reviewer #2: This protocol describes a well-designed, triple-masked, randomized trial that addresses an important, clinically relevant question. The primary endpoint and sample size calculation are clearly specified and methodologically appropriate. The study's strengths include rigorous blinding, validated delirium assessments, and trial registration. However, the expected effect size may be overly optimistic, and the heterogeneity of surgical populations warrants further consideration. The protocol would be strengthened by clarifying multiplicity handling and analytical details.

MAJOR COMMENTS

Clarity and Appropriateness of Endpoints:

The primary endpoint, the incidence of postoperative delirium within seven days or until discharge (assessed twice daily using the 3D-CAM), is clearly defined, clinically relevant, and based on a validated instrument.

Secondary and exploratory endpoints are clearly listed. However, the manuscript includes a large number of these outcomes, which increases the risk of multiplicity and selective emphasis. The authors should clarify which secondary outcomes are considered key and explain how they will handle multiplicity (e.g., hierarchical testing or interpretation as hypothesis-generating).

No information is provided about the effect of neostigmine and sugammadex on neuromuscular block reversal. This could be another important secondary endpoint that most readers would expect.

Consistency between endpoints and sample size calculation:

The sample size calculation is explicitly based on the primary endpoint (POCD in the first 7 days) and is methodologically sound, with clearly stated assumptions (baseline POD incidence, expected absolute risk reduction, α, and power).

However, the assumed 12.5% absolute risk reduction is derived from early cognitive dysfunction (first POD) rather than POD itself. The authors should better justify the clinical plausibility of this effect size for delirium or acknowledge the risk of overestimating the expected treatment effect.

Study Design and Internal Validity:

The randomized, triple-masked design with standardized anesthetic management and quantitative neuromuscular monitoring is a major strength.

Nonetheless, the inclusion of a heterogeneous surgical population (thoracic, abdominal, orthopedic, spinal, and urologic surgeries) may dilute the effect size and introduce clinical heterogeneity that could affect POD incidence. This issue should be discussed more explicitly, and stratification or interaction analyses should be specified more clearly in advance.

Planned Statistical Analysis:

The statistical plan is generally appropriate for the primary endpoint.

The use of the Kaplan-Meier method to analyze time to POD should be more clearly justified, especially with regard to how discharges before day 7 will be handled analytically (competing risk versus censoring).

No plan for handling missing data is provided beyond stating that no imputation will be used, which may be problematic in an elderly population with repeated assessments.

Discussion and Interpretation:

As this is a protocol, the discussion appropriately focuses on rationale and mechanisms. However, some statements anticipate benefits and clinical impacts that may appear overly optimistic for a superiority trial whose results are not yet known. The discussion should more explicitly acknowledge uncertainty and the possibility of neutral or negative findings.

Ethics Approval and Trial Registration:

Ethics approval is clearly reported with the committee name and approval number. Trial registration in the Chinese Clinical Trial Registry is appropriately reported, including the registration number and the date prior to patient enrollment.

These aspects are adequately addressed and compliant with international standards.

MINOR COMMENTS

Terminology: Ensure consistent use of "postoperative delirium (POD)" throughout the manuscript. Avoid repeating full definitions once they have been established.

Exploratory Outcomes: The long list of exploratory outcomes (e.g., pain, QoR-15, complications, and mortality) could be streamlined or clearly framed as descriptive only.

Generalizability: Excluding ICU patients and cardiac surgery patients limits the study's external validity, and this should be more clearly acknowledged as a design choice.

A brief description of TOF technical data is warranted, including the type of monitoring device, whether the calibration protocol is prespecified, and the choice of monitoring site.

.

Reviewer #1: **Yes:**Yahya Kayed AbuJwaidYahya Kayed AbuJwaidYahya Kayed AbuJwaidYahya Kayed AbuJwaid

Reviewer #2: **Yes:**Federico PiccioniFederico PiccioniFederico PiccioniFederico Piccioni

---

## [Author Response · Author response to Decision Letter 1]

1 Feb 2026

Additional Editor Comments:

This protocol describes a well-designed, triple-masked, randomized trial with a clearly defined primary endpoint. Key methodological concerns requiring clarification before implementation include: the handling of missing outcome data, which currently aligns with a modified intention-to-treat approach and risks bias if missing non-randomly;

Response: We appreciate this important suggestion. Our recently completed pilot trial (Long et al., Anesth Analg 2024, PMID: 37874773) employed an identical assessment protocol (twice-daily 3D-CAM evaluations by the same trained research team at our institution) in a comparable elderly surgical population (major non-cardiac surgery, mean age 69.6 years). In that study, POD assessments were completed without exception in all 78 randomized patients (100% ascertainment). Based on this, we expect minimal missing data (< 5%) in the current trial and have retained the complete case analysis as primary. Comprehensive sensitivity analyses will also be conducted to address any unforeseen missingness.

We have clarified this in the Statistical Analysis: “The primary analysis will be conducted in a modified intention-to-treat population, defined as all randomized patients who undergo surgery and have at least one postoperative POD assessment. In our recently completed pilot trial using identical assessment protocols, twice-daily 3D-CAM evaluations were completed without exception in all 78 older surgical patients (100% ascertainment). Based on this empirical evidence, we expect minimal missing data (< 5%) and will perform the primary analysis on a complete case basis. To assess robustness to any unforeseen missingness, we will conduct: (1) worst-case scenario analysis (all missing data assumed to be POD in the neostigmine/glycopyrrolate group and no POD in the sugammadex group); (2) best-case scenario analysis (reverse of worst-case); and (3) per-protocol analysis (excluding patients with major protocol deviations). If the observed missing data rate exceeds 10%, an additional sensitivity analysis using multiple imputation will be conducted.” (see Methods, Statistical analysis, lines 218‒230)

the detailed statistical specification for the time-to-event analysis of POD, including the handling of interval-censored data and discharges;

Response: Thank you. We have clarified the Kaplan-Meier method to analyze time to POD, including the survival model, time origin, and assumptions: “Kaplan–Meier curves will be used to describe time to first POD, with the day of surgery set as day 0. Time-to-event analysis will be conducted on a discrete-time scale (calendar days). The treatment effect will be estimated with a Cox proportional-hazards model. As a sensitivity analysis for competing risks (death and early discharge), we will estimate cumulative incidence functions using the Fine-Gray model. If the proportional hazards assumption is violated (Schoenfeld residuals test P < 0.05), we will use a time-dependent covariate model or stratified Cox model.” (see Methods, Statistical analysis, lines 232-238)

the potential for differential post-reversal propofol exposure between groups as a confounder;

Response: Thank you. We acknowledge the concern regarding potential confounding from differential post-reversal propofol exposure. To address this: (1) We have clarified that propofol sedation is standardized to BIS 60–80 until TOF ratio > 0.9 during the post-reversal period in both groups, minimizing discretionary differences: “Patients will be sedated with propofol infusion of 1–3 mg/kg/h to achieve a BIS of 60–80 until TOF ratio > 0.9.” (see Methods, Study interventions, lines 134‒136)

(2) We will record and compare cumulative propofol infusion dose between groups: “Cumulative propofol dose from study drug administration (reversal) to extubation will be recorded.” (see Methods, Data collection and management, lines 184‒185)

(3) Propofol exposure will be included as a covariate in sensitivity analysis to assess its confounding effect: “In addition, we will adjust for propofol exposure as a covariate in the sensitivity analysis of the primary endpoint.” (see Methods, Statistical analysis, lines 230‒231)

and the risk of differential outcome ascertainment due to variable discharge timing, necessitating pre-specified sensitivity analyses.

Response: Thank you for this important comment. We have acknowledged this potential ascertainment bias and clarified that this limitation will be addressed through analytical approaches: “The primary outcome window (within 7 postoperative days or until discharge) may lead to differential outcome ascertainment if discharge timing differs between groups. We will conduct a sensitivity landmark analysis restricted to patients with a minimum observation time of 72 hours (i.e., those remaining hospitalized for ≥ 3 days), ensuring comparable ascertainment windows between groups. In our pilot study, the median postoperative hospital stay was 7–10 days with only 2 patients (< 3%) discharged before postoperative day 3, indicating that this sensitivity analysis will retain > 97% of the cohort while effectively addressing potential ascertainment bias.” (see Methods, Statistical analysis, lines 247‒255)

The sample size calculation, while explicit, would benefit from a stronger justification for the assumed effect size on delirium.

Response: Thank you. We have clarified the sample size calculation and provided additional justification based on our internal pilot data: “We conservatively assumed a 12.5% absolute risk reduction (25% with sugammadex vs. 12.5% with neostigmine/glycopyrrolate), which is supported by our pilot trial conducted at the same center using identical 3D-CAM assessments. In that study, active interventions showed approximately 11% absolute reductions in POD incidence. This empirical evidence, combined with the anti-inflammatory and anti-oxidative properties of enhanced cholinergic signaling by neostigmine, supports the clinical plausibility of the assumed effect size.” (see Methods, Sample size calculation, lines 203‒208)

The plan should also address multiplicity given the numerous secondary outcomes, specify handling of clinical heterogeneity from the mixed surgical population, and provide more detail on quantitative neuromuscular monitoring.

Response: We appreciate these important methodological suggestions. We have clarified the multiplicity control for the secondary and exploratory endpoints: “To control for multiplicity, we will apply the Bonferroni correction for three secondary outcomes (days with POD, highest CAM-S score, and 30-day cognitive function), using a significance threshold of α = 0.017 (0.05/3). All exploratory outcomes (including pain scores, QoR-15 scores, non-delirium complications, 30-day mortality, and neuromuscular recovery metrics) will be analyzed using descriptive statistics only and interpreted as hypothesis-generating without adjustment for multiple comparisons.” (see Methods, Statistical Analysis, lines 256‒262)

While the primary objective is cognitive protection, we agree that documenting neuromuscular recovery is essential. We have added: “In addition, the neuromuscular recovery outcomes include time to TOF ratio > 0.9, incidence of residual neuromuscular blockade at extubation (TOF ratio < 0.9), time to extubation, and length of PACU stay.” (see Methods, Study outcomes, lines 168‒171)

Strengths include standardized anesthesia, rigorous masking, and proper registration. Minor revisions suggested for improved reproducibility and clarity involve detailing assessor training, specifying data sharing plans, and streamlining terminology.

Response: Thank you for this suggestion. We have added details regarding assessor training and inter-rater reliability in the Methods: “Postoperative follow-up will be performed by two independent outcome assessors, who will undergo standardized training prior to study initiation (including completion of the online 3D-CAM training and bedside practical training supervised by a board-certified neuropsychologist), as specified in our recent pilot study. Inter-rater reliability will be assessed during the initial phase of the trial (first 20 patients) using Cohen's kappa; an acceptable value of ≥ 0.80 will be required before formal recruitment proceeds.” (see Methods, Data collection and management, lines 185‒192)

We have clarified the data availability statement: “All de-identified individual participant data collected during the trial, study protocol, statistical analysis plan, and informed consent form, will be made available in the Mendeley Data at the time of publication. Identifiers (e.g., name, medical record number, exact date of birth, contact information) will be removed and replaced with unique study IDs.” (see Data availability statement, lines 330‒334)

We have ensured consistent use of “postoperative delirium (POD)” throughout the manuscript, with the full term defined only at first mention in the Abstract and Introduction, and the abbreviation “POD” used uniformly thereafter.

Reviewer #1: This manuscript presents a well-structured study protocol for a single-center, triple-masked randomized controlled trial comparing neostigmine/glycopyrrolate with sugammadex for neuromuscular blockade reversal, with postoperative delirium (POD) as the primary outcome in older adults. The clinical rationale is relevant, the research question is clearly stated, and the protocol includes ethical approval and trial registration. Overall, the study has the potential to address an important perioperative question; however, several methodological and statistical aspects require clarification to strengthen internal validity and replicability.

Major comments:

The analysis population is described as including randomized patients with available outcome data, and no imputation is planned for missing data. This corresponds to a modified intention-to-treat approach and may introduce bias if POD assessments are missing non-randomly (e.g., early discharge or clinical deterioration). The authors should clearly define the primary analysis set (ITT vs modified ITT) and pre-specify how missing POD assessments will be handled, including consideration of sensitivity analyses.

Response: We appreciate the reviewer’s concern regarding potential bias from missing data. Importantly, our recently completed pilot trial (Long et al., Anesth Analg 2024, PMID: 37874773) employed an identical assessment protocol (twice-daily 3D-CAM evaluations by the same trained research team at our institution) in a comparable elderly surgical population (major non-cardiac surgery, mean age 69.6 years). In that study, POD assessments were completed without exception in all 78 randomized patients (100% ascertainment). Based on this, we expect minimal missing data (< 5%) in the current trial and have retained the complete case analysis as primary. Comprehensive sensitivity analyses will also be conducted to address any unforeseen missingness.

We have clarified this in the Statistical Analysis: “The primary analysis will be conducted in a modified intention-to-treat population, defined as all randomized patients who undergo surgery and have at least one postoperative POD assessment. In our recently completed pilot trial using identical assessment protocols, twice-daily 3D-CAM evaluations were completed without exception in all 78 older surgical patients (100% ascertainment). Based on this empirical evidence, we expect minimal missing data (< 5%) and will perform the primary analysis on a complete case basis. To assess robustness to any unforeseen missingness, we will conduct: (1) worst-case scenario analysis (all missing data assumed to be POD in the neostigmine/glycopyrrolate group and no POD in the sugammadex group); (2) best-case scenario analysis (reverse of worst-case); and (3) per-protocol analysis (excluding patients with major protocol deviations). If the observed missing data rate exceeds 10%, an additional sensitivity analysis using multiple imputation will be conducted.” (see Methods, Statistical analysis, lines 218‒230)

The statistical plan mentions Kaplan–Meier analysis and hazard ratios for time to POD, but does not specify the survival model, time origin, or how event time will be assigned given twice-daily assessments (with potential interval censoring). These elements should be explicitly defined, and assumptions (e.g., proportional hazards) should be addressed.

Response: Thank you for this helpful comment. We have clarified the assessment of POD, “In addition to the twice-daily POD assessments, a 12-h look-back data extraction from medical records and nursing notes will be performed to capture any CAM-positive signs that may have occurred within the preceding 12 hours. The day of the first positive 3D-CAM assessment (morning, evening, or 12-h look-back) is taken as the event date.” (see Methods, Study outcomes, lines 147‒152)

We have clarified the survival model, time origin, and assumptions in the Statistical Analysis: “Kaplan–Meier curves will be used to describe time to first POD, with the day of surgery set as day 0. Time-to-event analysis will be conducted on a discrete-time scale (calendar days). The treatment effect will be estimated with a Cox proportional-hazards model. As a sensitivity analysis for competing risks (death and early discharge), we will estimate cumulative incidence functions using the Fine-Gray model. If the proportional hazards assumption is violated (Schoenfeld residuals test P < 0.05), we will use a time-dependent covariate model or stratified Cox model.” (see Methods, Statistical analysis, lines 232‒238)

The primary outcome window (“within 7 postoperative days or until discharge”) may lead to differential outcome ascertainment if discharge timing differs between groups. The authors should clarify how this limitation will be addressed analytically and consider sensitivity analyses based on minimum observation time.

Response: Thank you for this important comment. We have acknowledged this potential ascertainment bias and clarified that this limitation will be addressed through analytical approaches: “The primary outcome window (within 7 postoperative days or until discharge) may lead to differential outcome ascertainment if discharge timing differs between groups. We will conduct a sensitivity landmark analysis restricted to patients with a minimum observation time of 72 hours (i.e., those remaining hospitalized for ≥ 3 days), ensuring comparable ascertainment windows between groups. In our pilot study, the median postoperative hospital stay was 7–10 days with only 2 patients (< 3%) discharged before postoperative day 3, indicating that this sensitivity analysis will retain > 97% of the cohort while effectively addressing potential ascertainment bias.” (see Methods, Statistical analysis, lines 247‒255)

Minor comments:

Additional details on assessor training and any assessment of inter-rater reliability for POD diagnosis would improve reproducibility.

Response: Thank you for this suggestion. We have added details regarding assessor training and inter-rater reliability in the Methods: “Postoperative follow-up will be performed by two independent outcome assessors, who will undergo standardized training prior to study initiation (including completion of the online 3D-CAM training and bedside practical training supervised by a board-certified neuropsychologist), as specified in our recent pilot study. Inter-rater reliability will be assessed during the initial phase of the trial (first 20 patients) using Cohen's kappa; an acceptable value of ≥ 0.80 will be required before formal recruitment proceeds.” (see Methods, Data collection and management, lines 185‒192)

Clarification on whether MMSE scores are adjusted for education or language norms would strengthen baseline cognitive characterization.

Response: Thank you. We have clarified the education-adjusted MMSE scoring in the Methods: “Baseline cognitive function will be assessed with the 30-point Chinese version of the Mini-Mental State Examination (MMSE), covering six domains: orientation to time and place, word repetition, attention and calculation, language expression, speech comprehension, and simple motor commands. E

---

## [Decision Letter · Decision Letter 1]

19 Feb 2026

Dear Dr. Peng,

Thank you for submitting your manuscript to PLOS ONE. After careful consideration, we feel that it has merit but does not fully meet PLOS ONE’s publication criteria as it currently stands. Therefore, we invite you to submit a revised version of the manuscript that addresses the points raised during the review process.

We look forward to receiving your revised manuscript.

Kind regards,

Lalit Gupta

Academic Editor

PLOS One

**Journal Requirements:**

**Additional Editor Comments:**

The revised protocol has been demonstrated to be both methodologically sound and substantively improved. It is this author's opinion that a few minor suggestions can be made to enhance transparency and clarity.

1. The missing data assumption posits that, while minimal missingness is anticipated, it may be beneficial to explicitly articulate the assumption underlying the primary complete-case analysis (e.g., missing completely at random) and briefly acknowledge this in the interpretation framework.

2. The time-to-event approximation is determined as follows: In the context of twice-daily POD assessments, event timing is interval-censored within a 12-hour window. A concise acknowledgment that the discrete-time approach constitutes an approximation would enhance methodological transparency.

3. The proposed plan to adjust for cumulative propofol exposure is deemed to be appropriate. It is imperative to consider the specific regression framework in which this covariate will be incorporated.

4. The assumption of effect size is predicated on the premise that, while substantiated by preliminary data, the anticipated absolute risk reduction of 12.5% may in fact prove to be overly optimistic. A brief acknowledgment that a smaller true effect could result in reduced power would serve to strengthen balance in the protocol.

5. Multiplicity and secondary outcomes – The Bonferroni approach is deemed appropriate. It is imperative to elucidate that secondary analyses will persist in their capacity to provide supportive evidence and to generate hypotheses, a state of affairs that may engender enhanced interpretive consistency.

6. Subgroup Analyses

As multiple subgroup and interaction analyses have been predetermined, elucidating their exploratory nature (absent a substantial interaction detection) would prove beneficial.

7. Neuromuscular Monitoring – The inclusion of supplementary technical details is appreciated. A brief statement confirming adherence to current international recommendations for quantitative neuromuscular monitoring would further strengthen reproducibility.

Reviewers' comments:

Reviewer's Responses to Questions

**Comments to the Author**

1. Does the manuscript provide a valid rationale for the proposed study, with clearly identified and justified research questions?

Reviewer #1: Yes

Reviewer #2: Yes

2. Is the protocol technically sound and planned in a manner that will lead to a meaningful outcome and allow testing the stated hypotheses?

Reviewer #1: Yes

Reviewer #2: Yes

3. Is the methodology feasible and described in sufficient detail to allow the work to be replicable?

Reviewer #1: Yes

Reviewer #2: Yes

4. Have the authors described where all data underlying the findings will be made available when the study is complete?

The PLOS Data policy requires authors to make all data underlying the findings described in their manuscript fully available without restriction, with rare exception, at the time of publication. The data should be provided as part of the manuscript or its supporting information, or deposited to a public repository. For example, in addition to summary statistics, the data points behind means, medians and variance measures should be available. If there are restrictions on publicly sharing data—e.g. participant privacy or use of data from a third party—those must be specified.requires authors to make all data underlying the findings described in their manuscript fully available without restriction, with rare exception, at the time of publication. The data should be provided as part of the manuscript or its supporting information, or deposited to a public repository. For example, in addition to summary statistics, the data points behind means, medians and variance measures should be available. If there are restrictions on publicly sharing data—e.g. participant privacy or use of data from a third party—those must be specified.

Reviewer #1: Yes

Reviewer #2: Yes

5. Is the manuscript presented in an intelligible fashion and written in standard English?

Reviewer #1: Yes

Reviewer #2: Yes

You may also provide optional suggestions and comments to authors that they might find helpful in planning their study.

Reviewer #1: The authors have satisfactorily addressed the methodological and statistical concerns raised in the previous review. The revised manuscript provides clarified definitions of the analysis population, pre-specified sensitivity analyses for missing data and competing risks, detailed time-to-event modeling assumptions, and improved transparency regarding anesthetic management and outcome assessment. The protocol is now methodologically sound and sufficiently specified to ensure internal validity and reproducibility. I have no further major concerns.

Reviewer #2: The revised protocol has been demonstrated to be both methodologically sound and substantively improved. It is this author's opinion that a few minor suggestions can be made to enhance transparency and clarity.

1. The missing data assumption posits that, while minimal missingness is anticipated, it may be beneficial to explicitly articulate the assumption underlying the primary complete-case analysis (e.g., missing completely at random) and briefly acknowledge this in the interpretation framework.

2. The time-to-event approximation is determined as follows: In the context of twice-daily POD assessments, event timing is interval-censored within a 12-hour window. A concise acknowledgment that the discrete-time approach constitutes an approximation would enhance methodological transparency.

3. The proposed plan to adjust for cumulative propofol exposure is deemed to be appropriate. It is imperative to consider the specific regression framework in which this covariate will be incorporated.

4. The assumption of effect size is predicated on the premise that, while substantiated by preliminary data, the anticipated absolute risk reduction of 12.5% may in fact prove to be overly optimistic. A brief acknowledgment that a smaller true effect could result in reduced power would serve to strengthen balance in the protocol.

5. Multiplicity and secondary outcomes – The Bonferroni approach is deemed appropriate. It is imperative to elucidate that secondary analyses will persist in their capacity to provide supportive evidence and to generate hypotheses, a state of affairs that may engender enhanced interpretive consistency.

6. Subgroup Analyses

As multiple subgroup and interaction analyses have been predetermined, elucidating their exploratory nature (absent a substantial interaction detection) would prove beneficial.

7. Neuromuscular Monitoring – The inclusion of supplementary technical details is appreciated. A brief statement confirming adherence to current international recommendations for quantitative neuromuscular monitoring would further strengthen reproducibility.

The protocol was meticulously formulated and succinctly documented. Addressing these minor points would further enhance methodological transparency and interpretative clarity.

.

Reviewer #1: **Yes:**Yahya Kayed AbuJwaidYahya Kayed AbuJwaidYahya Kayed AbuJwaidYahya Kayed AbuJwaid

Reviewer #2: No

---

## [Author Response · Author response to Decision Letter 2]

27 Feb 2026

1. The missing data assumption posits that, while minimal missingness is anticipated, it may be beneficial to explicitly articulate the assumption underlying the primary complete-case analysis (e.g., missing completely at random) and briefly acknowledge this in the interpretation framework.

Thank you. We have now explicitly stated in the Statistical analysis section that the primary complete-case analysis assumes data are missing completely at random, given the anticipated minimal missingness (< 5%) based on our pilot trial experience. (see page 10, lines 232‒233)

We have also added a brief acknowledgment in the Discussion regarding this assumption and its potential limitation. (see page 15, lines 336‒340)

2. The time-to-event approximation is determined as follows: In the context of twice-daily POD assessments, event timing is interval-censored within a 12-hour window. A concise acknowledgment that the discrete-time approach constitutes an approximation would enhance methodological transparency.

Thank you. We have now explicitly acknowledged in the Statistical analysis section that the discrete-time approach for time-to-event analysis constitutes an approximation, given that the twice-daily POD assessments result in interval-censored event timing within a 12-hour window. (see page 10, lines 242‒243)

3. The proposed plan to adjust for cumulative propofol exposure is deemed to be appropriate. It is imperative to consider the specific regression framework in which this covariate will be incorporated.

Thank you. We have now clarified that cumulative propofol exposure will be incorporated as a continuous covariate in the logistic regression model for the primary binary outcome and in the Cox proportional-hazards model for time-to-event analysis, with effect estimates reported as adjusted odds ratio and hazard ratio, respectively. (see page 11, lines 248‒252)

4. The assumption of effect size is predicated on the premise that, while substantiated by preliminary data, the anticipated absolute risk reduction of 12.5% may in fact prove to be overly optimistic. A brief acknowledgment that a smaller true effect could result in reduced power would serve to strengthen balance in the protocol.

Thank you. We have now added a brief acknowledgment in the Sample size calculation section that the assumed 12.5% absolute risk reduction, while supported by preliminary data and mechanistic rationale, may prove optimistic, and that a smaller true effect would result in reduced statistical power. (see page 9, lines 214‒217)

5. Multiplicity and secondary outcomes – The Bonferroni approach is deemed appropriate. It is imperative to elucidate that secondary analyses will persist in their capacity to provide supportive evidence and to generate hypotheses, a state of affairs that may engender enhanced interpretive consistency.

Thank you. We have now clarified in the Statistical analysis section that secondary analyses will serve a supportive role for the primary outcome and help generate hypotheses for future research, ensuring balanced interpretation of findings. (see page 12, lines 274‒276)

6. Subgroup Analyses

As multiple subgroup and interaction analyses have been predetermined, elucidating their exploratory nature (absent a substantial interaction detection) would prove beneficial.

Thank you. We have now explicitly stated in the Statistical analysis section that all subgroup and interaction analyses are exploratory in nature unless a substantial interaction is detected, to prevent overinterpretation of subgroup findings. (see page 11, lines 258‒260)

7. Neuromuscular Monitoring – The inclusion of supplementary technical details is appreciated. A brief statement confirming adherence to current international recommendations for quantitative neuromuscular monitoring would further strengthen reproducibility.

Thank you. We have now added a brief statement in the Anesthetic care section confirming that our neuromuscular monitoring protocol adheres to current international recommendations for quantitative monitoring (ESAIC 2023 and ASA 2023 guidelines), ensuring standardized calibration and assessment of neuromuscular block and recovery. (see page 5, lines 116‒120)

---

## [Decision Letter · Decision Letter 2]

19 Mar 2026

Effect of neostigmine/glycopyrrolate versus sugammadex on postoperative delirium in older adults: a triple-masked, randomized, controlled trial protocol

PONE-D-25-66069R2

Dear Dr. Peng,

We’re pleased to inform you that your manuscript has been judged scientifically suitable for publication and will be formally accepted for publication once it meets all outstanding technical requirements.

Kind regards,

Lalit Gupta

Academic Editor

PLOS One

Additional Editor Comments (optional):

Reviewers' comments:

Reviewer's Responses to Questions

**Comments to the Author**

1. Does the manuscript provide a valid rationale for the proposed study, with clearly identified and justified research questions?

Reviewer #2: Yes

2. Is the protocol technically sound and planned in a manner that will lead to a meaningful outcome and allow testing the stated hypotheses?

Reviewer #2: Yes

3. Is the methodology feasible and described in sufficient detail to allow the work to be replicable?

Reviewer #2: Yes

4. Have the authors described where all data underlying the findings will be made available when the study is complete?

The PLOS Data policy requires authors to make all data underlying the findings described in their manuscript fully available without restriction, with rare exception, at the time of publication. The data should be provided as part of the manuscript or its supporting information, or deposited to a public repository. For example, in addition to summary statistics, the data points behind means, medians and variance measures should be available. If there are restrictions on publicly sharing data—e.g. participant privacy or use of data from a third party—those must be specified.requires authors to make all data underlying the findings described in their manuscript fully available without restriction, with rare exception, at the time of publication. The data should be provided as part of the manuscript or its supporting information, or deposited to a public repository. For example, in addition to summary statistics, the data points behind means, medians and variance measures should be available. If there are restrictions on publicly sharing data—e.g. participant privacy or use of data from a third party—those must be specified.

Reviewer #2: Yes

5. Is the manuscript presented in an intelligible fashion and written in standard English?

Reviewer #2: Yes

You may also provide optional suggestions and comments to authors that they might find helpful in planning their study.

Reviewer #2: I would like to thank the authors for their careful and constructive revision of the manuscript. The responses to the reviewers’ comments are clear and appropriate, and the manuscript has been improved accordingly.

.

Reviewer #2: No
